# Surface Roughness of Varnished Wood Pre-Treated Using Sanding and Thermal Compression

Pavlo Bekhta [1,*] , Barbara Lis [2], Tomasz Krystofiak [2] and Nataliya Bekhta [3]

1. Department of Wood-Based Composites, Cellulose and Paper, Ukrainian National Forestry University, 79057 Lviv, Ukraine
2. Department of Wood Science and Thermal Technics, Poznań University of Life Sciences, 60-627 Poznań, Poland; barbara.lis@up.poznan.pl (B.L.); tomasz.krystofiak@up.poznan.pl (T.K.)
3. Department of Design, Ukrainian National Forestry University, 79057 Lviv, Ukraine; n.bekhta@nltu.edu.ua
* Correspondence: bekhta@nltu.edu.ua

**Abstract:** Surface roughness is an important factor during the processes of wood gluing and finishing. This study proposed a new approach for the preparation of wood veneer surfaces before varnishing through the use of thermal compression instead of sanding. The quality of the pre-treated surface was examined using surface roughness measurements. In the experiment, a wood veneer of black alder and birch, before varnishing, was subjected to sanding with a sandpaper of 180 grit size, and thermal compression at temperatures of 180 and 210 °C. Three different types of commercially manufactured varnishes (water-based (WB), polyurethane (PUR) and UV-cured (UV)) were applied to the prepared veneer surfaces with various numbers of varnish layers. Seven roughness parameters such as $R_a$, $R_z$, $R_q$, $R_p$, $R_v$, $R_{sk}$, and $R_{ku}$ were determined for the sanded and thermally densified unvarnished and varnished surfaces. The profile surface was recorded with a portable surface roughness tester along and across the wood fibers. It was found that there was no difference between the surface roughnesses of the surfaces that had been sanded and the surface roughnesses of those that had been thermally densified at a temperature of 210 °C. The research suggests that thermal compression at a temperature of 210 °C is enough to obtain smoother surfaces with a UV varnish system, and this process can be recommended as a replacement for sanding before varnishing as the most labor-intensive and expensive operations in woodworking industry. Applying two layers of varnish along with intermediate sanding was also sufficient to obtain a satisfactory finish.

**Keywords:** surface roughness; sanding; thermal compression; varnish system; wood veneer

## 1. Introduction

The concept of roughness is a very important issue in any manufacturing process of wood/wood-based materials, as most of these materials are obtained in the process of gluing. In addition, almost all of the wood materials are subjected to the finishing process. The surface roughness of the wood plays a very important role in both these processes of gluing and finishing. Surface roughness affects not only the formation of the surface quality of the final products, but also the properties of the adhesive and the varnish applied, as well as their costs and the parameters of pressing and finishing. It is well known that the higher the surface roughness, the greater the consumption of glue [1,2]. The glue should be sufficient to fill the valleys and to form a continuous adhesive layer of equal thickness. Rough veneer requires a higher pressing pressure and it reduces the contact between the layers, which leads to weak adhesion strength and the poorer properties of plywood [3]. Faust and Rice [4] found that a rough veneer can reduce the adhesion strength by 33% compared to a smooth veneer. A rough veneer can also cause excessive resin use, and can lead to resin bleeding through the face veneer [1,2], increasing production costs. The same applies to the lamination of particleboard and medium-density fiberboard with wood veneer in the furniture industry [5], as well as the application of varnish on wood

surfaces. Raabe et al. [6] mentioned that the success of the surface finishing of the final product is closely linked to the wood surface smoothness. Therefore, the smoothness of the surface is very important for using lesser amounts of surface finishing materials [7]. Smooth surfaces need a relatively low quantity of paint for coverage, and they show the best paint performances, even on low-grade wood [8]. The relationship between surface roughness and some properties of veneer-based products is well known [3,4,8–12].

To improve the appearance and color of wood panels, a sliced decorative veneer with a low thickness of 0.2–0.9 mm as an overlay material made of naturally beautiful precious woods is usually employed [13]. However, part of the precious wood is irretrievably lost during the sanding of the veneered wood panels before finishing [14]. In addition, the resources of valuable wood species, which can be used as a material for veneering, are rapidly depleting and becoming more expensive. Therefore, there is a need to look for opportunities to replace high-value wood species with low-grade ones. Less valuable wood species, such as black alder and birch, are characterized by various defects, mechanical damage, and poorer decorative appearance than valuable wood species. Therefore, they are often considered only as low-grade wood species, a secondary resource that is mainly used for the production of plywood, joinery and packaging, and energy. This necessitates the improvement of the aesthetic properties of the veneers of such wood species. On the other hand, Bulian and Graystone [15] argue that varnishes can add value to low-value wood or to those used less commonly in the market to give them a desired aesthetic appearance.

However, the final quality of the finish depends on several elements, including the application of the coating method, the substrate characteristics such as roughness, the chemical structure, and the interaction between the coating and the substrate [16]. The preparation of the surface of the wood substrate is one of the most important processes before finishing and it has a significant impact on the surface properties of the coating, including color, gloss, and roughness, as well as the adhesion of the coating to the substrate. Usually this process can be performed via helical planing, face milling, and sanding, etc. [17]. Sanding is the most common pre-treatment process of a wooden surface before finishing. The process makes the surface more homogeneous and smooth [8,17], which is a prerequisite for quality interaction between the coating and the substrate, and it is also necessary for a good appearance [18]. A significant disadvantage is that during the sanding, a layer of valuable wood species is usually removed, and this portion of wood irreversibly goes to waste, generating economic losses. In addition, sander dust creates unfavorable conditions for workers and pollutes the environment. Moreover, sanding is one of the most qualified, time-consuming, and expensive operations in the woodworking industry [19].

On the other hand, in our previous studies [20–25], it was found that the combination of heat treatment and compression improves the overall surface quality of the samples, making it denser, smoother, and homogeneous. After such a treatment, sanding of the compressed wood surface before finishing is no longer required [14,22]. This process is called thermo-mechanical densification, and it is the simplest and most environmentally friendly method of wood modification, as it does not use chemicals [26,27]. Thermo-mechanical densification makes it possible for new and useful characteristics to be provided to wood species that are of low quality and technical characteristics [20–25]. In particular, in addition to the benefits to the wood properties, such as strength, surface hardness, and durability [5,28], the surface quality, and in particular, the aesthetic properties, can also be improved [5,28]; the color of wood becomes more attractive [20,29], the surface roughness decreases [22,30–32], and the surface becomes glossier and smoother [21,22,32], while the need for sanding is minimized. This improved attractiveness of the veneer surface facilitates the application of transparent organic coatings that allow for improvements to the natural characteristics of wood to remain visible, and so the demand for them has been increasing. Therefore, densification is a method of utilizing low-density wood species instead of high density species in applications of higher value [33]. In addition, after such treatments, wood species with lower quality decorative characteristics are made in a color that is similar to "exotic wood" [20]. Additionally, transparent coatings allow for the

acquired good aesthetic properties of the wood to be kept. Together, this can increase the demand for such wood/wood-based panels, as well as the determination of their value and price.

Therefore, it was proposed that the time-consuming process of sanding the wooden substrate before finishing with a thermal compression pre-treatment be replaced [34]. However, little or no information is available on the possibility for using thermally compressed wood as a substrate for finishing [14,35–38]. Much more information regarding the finishing of heat-treated wood can be found in [39–43]. At the same time, the replacement of the sanding process with the thermal compression pre-treatment of the wooden substrate makes it possible to produce veneered panels with an improved aesthetic appearance, a high adhesive strength, and lower varnish consumption [14,35]. The findings obtained in our previous study indicated that the "thermal compression of wood veneer followed by an appropriate transparent varnish system, could be considered as an industrially acceptable method to protect wood against photo-degradation in indoor conditions with simultaneous improvement of aesthetic surface properties and preservation of wood in the absence of sanding process" [36].

The quality of the surface of wood/wood-based panels is usually assessed via surface roughness [44], which affects the adhesion strength and the properties of the coating. Currently, there is enough literature data regarding the roughness of solid wood [17] and thermally densified wood [5,22,28,30–32]. However, there is presently no comprehensive information on the comparison of the roughness of the wooden sanded surface and the thermally densified surface. In addition, there is no information on the effect of sanding and thermal densification on the surface roughness of unvarnished and varnished black alder and birch woods.

Thus, the main objective of this work was to investigate the potential impact of a new pre-treatment process of the wooden surface substrate before finishing via thermal compression on the surface roughness. The surface roughness of three different varnish systems (WB, PUR, and UV) applied on MDF panel veneered with the sanded and thermally compressed wood veneer of black alder and birch with various numbers of varnish layers was evaluated. Sanding was used as a conventional surface pre-treatment process for the comparison.

## 2. Materials and Methods

### 2.1. Materials, the Pre-Treatment Process of the Wood Veneer, the Surface Varnishing Process, and Statistical Analysis

The content of this paragraph is described with sufficient details in our previous article [36]. Defect-free samples of veneers were purchased from the LLC "ODEK" company in Ukraine. The veneer samples was thermally compressed between the smooth and carefully cleaned heated plates of an open-system laboratory press at temperatures of 180 °C (TC-180) and 210 °C (TC-210) under a constant pressure of 2 MPa for a 3 min time span.

### 2.2. Surface Roughness Measurement

The surface roughnesses of natural alder and birch veneer were determined after thermal compression and sanding, as well as after varnishing with various varnish systems. The surface roughness measurements of all samples were recorded on the surface of the veneer samples before and after relevant treatment with a Mitutoyo Surftest SJ-210 Series 178 Portable Surface Roughness Tester, according to ISO 4287 [45]. All measurement results were processed using a digital Gaussian filter. The measurement error of unevenness did not exceed ±10%. Five measurements were taken from the surface of each sample, five along ($\parallel$) and five perpendicular ($\perp$) to the wood fibers. The following parameters of wood surface roughness were evaluated: arithmetic average height ($R_a$), average peak-to-valley roughness ($R_z$), root mean square ($R_q$), maximum peak height ($R_p$), maximum valley depth ($R_v$), skewness ($R_{sk}$), and kurtosis ($R_{ku}$).

## 3. Results

### 3.1. Surface Roughness of Sanded and Thermally Densified Samples

ANOVA analysis showed that the wood species has a negligible effect, and that the direction of the wood fibers and the method of the surface pre-treatment before varnishing significantly affected the parameters of the wood surface roughness $R_a$, $R_z$, $R_q$, $R_p$, $R_v$, $R_{sk}$, and $R_{ku}$. In addition, the method of pre-treatment of the wood substrate had a stronger effect on $R_a$, $R_q$, and $R_p$, while $R_z$, $R_v$, $R_{sk}$, and $R_{ku}$ were more dependent upon the direction of the wood fibers (Table 1). In this study, alder and birch wood did not differ significantly in terms of the surface roughness parameters (except $R_{ku}$).

**Table 1.** Analysis of variance of surface roughness for sanded and thermally compressed veneers.

| Source of Variation | *F* Value | | | | | | |
|---|---|---|---|---|---|---|---|
| | $R_a$ | $R_q$ | $R_z$ | $R_p$ | $R_v$ | $R_{sk}$ | $R_{ku}$ |
| Wood species (WS) | 0.260 ** | 0.013 ** | 0.303 ** | 0.117 ** | 0.580 ** | 0.018 ** | 6.794 * |
| Direction (D) | 56.841 * | 80.645 * | 231.559 * | 16.145 * | 394.654 * | 16.817 * | 62.409 * |
| Pre-treatment (PT) | 130.857 * | 138.102 * | 175.673 * | 95.056 * | 124.399 * | 8.063 * | 29.480 * |
| WS × D | 1.080 ** | 0.726 ** | 0.073 ** | 0.098 ** | 0.351 ** | 0.833 ** | 4.613 * |
| WS × PT | 1.715 ** | 1.596 ** | 1.513 ** | 0.506 ** | 1.994 ** | 1.148 ** | 1.182 ** |
| D × PT | 4.902 * | 4.447 * | 5.688 * | 4.080 * | 4.298 * | 1.335 ** | 9.889 * |
| WS × D × PT | 1.558 ** | 2.126 ** | 3.954 * | 0.388 ** | 6.344 * | 0.077 ** | 1.357 ** |

*: Significant ($p \leq 0.05$); **: Non-significant.

The surface roughness values measured along the wood fibers were less than the values measured across the fibers (Table 2). In particular, the roughness values along the wood fibers after sanding were 1.8–2.6 and 1.5–2.5 times lower compared to the roughness values across the fibers, respectively, for alder and birch woods. Vitosyte et al. [46] also reported that compared to the parameters obtained along the fibers, the surface roughness parameters of ash wood across the fibers were higher (depending on the grit size of the abrasive paper) $R_a$—up to 2.8, $R_z$—up to 3.6, and $R_{max}$—up to 5.2 times. Several other authors [7,47–49] also found that sanded woods with higher grit numbers presented with less roughness. The difference in the values of surface roughness along and across the fibers after thermal densification was smaller than after sanding. The roughness values of thermally densified wood along the fibers were 0.8–2.6 and 0.9–2.1 times lower than the roughness values across the fibers for non-densified alder and birch wood, respectively (Table 2). This difference in the values of roughness along and across the wood fibers is explained by the anatomical structure of the wood. Similar results were obtained by other authors [22,32] who investigated the effects of temperature and pressure of thermal densification on the surface roughnesses of different wood species.

The method of pre-treatment of the substrate surface before varnishing significantly affects the surface roughness. The lowest values of roughness were observed after sanding and TC-210. The important fact is that the ANOVA analysis showed that the surfaces of alder and birch woods after sanding and TC-210 differed insignificantly ($p > 0.05$) from each other in terms of the roughness parameters $R_a$, $R_q$, and $R_p$. A significant difference ($p \leq 0.05$) between these surfaces existed in the values of the roughness parameters $R_z$, $R_v$, $R_{sk}$, and $R_{ku}$. In particular, the sanded surface had 1.4, 1.6, 2.2, and 1.6 times, respectively, lower average values of $R_z$, $R_v$, $R_{sk}$, and $R_{ku}$ compared to the surface treated with TC-210. The values of the roughness parameters $R_a$, $R_q$, $R_z$, $R_p$, $R_v$, and $R_{sk}$ differed insignificantly for the surfaces after treatments TC-210 and TC-180. The comparison of the roughness parameters along and across the fibers of alder wood showed that the sanded surfaces and TC-210 differed significantly for the values of the parameters $R_a(\|)$, $R_q(\|)$, $R_z(\|)$, $R_v(\|)$, $R_{sk}(\|)$, $R_{ku}(\|)$, $R_{sk}(\perp)$, and $R_{ku}(\perp)$, but that they differed insignificantly for the parameters $R_p(\|)$, $R_a(\perp)$, $R_q(\perp)$, $R_z(\perp)$, $R_p(\perp)$, and $R_v(\perp)$. The sanded surfaces and TC-210 of birch wood differed significantly in the values of the parameters $R_v(\|)$, $R_z(\perp)$, $R_v(\perp)$, and $R_{ku}(\perp)$, but

they differed insignificantly in the parameters $R_a(\parallel)$, $R_q(\parallel)$, $R_z(\parallel)$, $R_p(\parallel)$, $R_{sk}(\parallel)$, $R_{ku}(\parallel)$, $R_a(\perp)$, $R_q(\perp)$, $R_p(\perp)$, and $R_{sk}(\perp)$.

**Table 2.** Average roughness values of the non-treated and pre-treated samples.

| Wood Species | Direction | Treatment | Roughness Parameters (µm) | | | | | | |
|---|---|---|---|---|---|---|---|---|---|
| | | | $R_a$ | $R_q$ | $R_z$ | $R_p$ | $R_v$ | $R_{sk}$ | $R_{ku}$ |
| Black alder | Along (∥) the fibers | NS | 11.96 ± 0.53 C [1] | 14.54 ± 1.01 C | 65.14 ± 9.58 C | 30.19 ± 2.92 B | 34.95 ± 6.74 C | −0.20 ± 0.26 B | 2.65 ± 0.44 A |
| | | Sanded | 3.81 ± 0.37 A | 4.77 ± 0.56 A | 22.24 ± 3.54 A | 10.84 ± 2.36 A | 11.40 ± 1.78 A | −0.05 ± 0.37 B | 2.93 ± 0.35 A |
| | | ND | 14.08 ± 0.60 D | 17.28 ± 0.45 D | 77.16 ± 1.86 D | 35.51 ± 1.43 D | 41.65 ± 2.69 D | −0.28 ± 0.15 B | 2.86 ± 0.38 A |
| | | TC-180 | 5.53 ± 1.06 B | 7.09 ± 1.23 B | 33.95 ± 3.85 B | 14.53 ± 3.60 A | 19.41 ± 1.62 B | −0.56 ± 0.49 AB | 3.91 ± 0.99 B |
| | | TC-210 | 5.06 ± 0.87 B | 6.58 ± 1.23 B | 32.62 ± 6.75 B | 12.33 ± 2.88 A | 20.29 ± 4.55 B | −0.86 ± 0.34 A | 4.11 ± 0.64 B |
| | Across (⊥) the fibers | NS | 21.68 ± 2.30 C | 26.85 ± 2.57 C | 130.99 ± 8.84 C | 50.28 ± 4.45 C | 80.70 ± 5.98 D | −0.56 ± 0.27 B | 2.97 ± 0.25 A |
| | | Sanded | 6.74 ± 0.72 A | 8.86 ± 1.14 A | 45.01 ± 6.26 A | 15.53 ± 1.65 A | 29.48 ± 5.07 A | −0.92 ± 0.19 B | 4.16 ± 0.56 AB |
| | | ND | 16.42 ± 5.29 B | 20.56 ± 6.15 B | 103.31 ± 22.80 B | 39.29 ± 11.16 B | 64.02 ± 12.25 C | −0.60 ± 0.24 B | 3.28 ± 0.54 A |
| | | TC-180 | 8.44 ± 1.68 A | 11.20 ± 1.92 A | 61.62 ± 6.55 A | 15.56 ± 2.66 A | 46.06 ± 4.29 B | −1.48 ± 0.29 A | 5.32 ± 1.19 B |
| | | TC-210 | 5.41 ± 1.98 A | 7.39 ± 2.35 A | 44.58 ± 9.90 A | 10.86 ± 4.38 A | 33.72 ± 6.33 A | −1.78 ± 0.48 A | 7.01 ± 2.14 C |
| Birch | Along (∥) the fibers | NS | 12.27 ± 0.74 C | 15.84 ± 1.08 C | 75.12 ± 5.90 C | 32.13 ± 4.64 B | 42.98 ± 1.26 D | −0.35 ± 0.26 B | 3.21 ± 0.07 AB |
| | | Sanded | 3.31 ± 0.96 A | 4.24 ± 1.21 A | 19.53 ± 4.45 A | 8.73 ± 2.05 A | 10.80 ± 2.65 A | −0.32 ± 0.28 B | 3.23 ± 0.57 AB |
| | | ND | 14.90 ± 1.18 D | 18.49 ± 1.56 D | 78.30 ± 6.95 C | 37.88 ± 4.41 C | 40.42 ± 3.69 D | −0.16 ± 0.18 B | 2.63 ± 0.23 A |
| | | TC-180 | 6.07 ± 1.80 B | 7.77 ± 2.18 B | 36.99 ± 9.08 B | 12.80 ± 3.54 A | 24.19 ± 6.17 C | −0.95 ± 0.46 A | 4.16 ± 1.25 B |
| | | TC-210 | 5.01 ± 1.45 AB | 6.32 ± 1.66 AB | 29.35 ± 6.97 AB | 11.69 ± 3.84 A | 17.65 ± 3.99 B | −0.69 ± 0.36 AB | 3.72 ± 0.97 AB |
| | Across (⊥) the fibers | NS | 16.93 ± 0.11 B | 21.53 ± 0.73 B | 108.80 ± 5.91 C | 44.25 ± 3.32 B | 64.56 ± 3.54 C | −0.31 ± 0.22 A | 3.10 ± 0.27 A |
| | | Sanded | 5.29 ± 0.94 A | 7.09 ± 1.24 A | 36.19 ± 4.76 A | 12.70 ± 3.52 A | 23.49 ± 3.48 A | −0.89 ± 0.67 A | 4.70 ± 0.76 A |
| | | ND | 18.07 ± 2.93 B | 22.89 ± 3.10 B | 114.57 ± 10.60 C | 42.60 ± 16.96 B | 65.98 ± 4.13 C | −0.20 ± 0.34 A | 3.33 ± 0.50 A |
| | | TC-180 | 7.53 ± 0.65 A | 10.69 ± 0.57 A | 64.03 ± 1.26 B | 15.16 ± 3.59 A | 48.87 ± 2.55 B | −1.86 ± 0.47 A | 7.76 ± 1.71 B |
| | | TC-210 | 6.88 ± 1.12 A | 9.86 ± 1.39 A | 60.07 ± 5.77 B | 11.87 ± 0.90 A | 48.20 ± 5.21 B | −1.22 ± 1.98 A | 8.75 ± 1.76 B |

[1] Averages followed by the same letter at the column are statistically equal according to the Duncan test at 95% probability.

In general, the obtained results confirmed our assumption that the labor-intensive operation of pre-treating the wood surface through sanding before varnishing could be replaced by the thermal compression process. This replacement, in addition to providing the necessary surface roughness for varnishing, makes this surface more attractive in terms of decorative properties [14,20,21,36]. Therefore, in terms of surface roughness and decorative properties, the recommended temperature for thermal compression could be 210 °C.

The values of the roughness parameters $R_a$, $R_q$, $R_z$, $R_p$, and $R_v$ of sanded alder wood were reduced compared to non-sanded wood by 3.1, 3.1, 2.9, 2.8, and 3.1 times, respectively, along the fibers, and by 3.2, 3.0, 2.9, 3.2, and 2.7 times, respectively, across the fibers. For the parameters $R_{sk}$ and $R_{ku}$, there was an increase in the values of roughness along the fibers by 4.0 and 1.1 times, respectively, and across the fibers, only $R_{ku}$ increased by 1.4 times, and $R_{sk}$ decreased by 1.6 times. For sanded birch wood, the decrease in the values of the roughness parameters $R_a$, $R_q$, $R_z$, $R_p$, and $R_v$ along the fibers was 3.7, 3.7, 3.8, 3.7, and 4.0 times, respectively, and across the fibers, they were 3.2, 3.0, 3.0, 3.5, and 2.7 times, respectively, compared with non-sanded wood. For the parameters $R_{sk}$ and $R_{ku}$, there was an increase in the roughness values by 2.8 and 1.1 times, and by 1.5 and 1.0 times, respectively, across and along the fibers. In another study [50], it was also shown that $R_a$ and $R_z$ were significantly higher across than along the fibers for all surfacing methods (helical planing, face milling, and sanding).

The temperature of the thermal compression significantly affects the surface roughness. With its increase, the surface roughness decreases. This is in good agreement with our previous studies [22,24], which showed that as the densification temperature increases, the surface roughness values decrease. Accordingly, in this study, for both of the studied wood species, the lowest roughness values along and across the fibers were observed at a densification temperature of 210 °C, which was higher than the temperature of 180 °C. The values of the roughness parameters $R_a$, $R_q$, $R_z$, $R_p$, $R_v$, and $R_{sk}$ for alder wood after treatment TC-210 were reduced by 2.8, 2.6, 2.4, 2.9, 2.1, and 3.1 times, respectively, along the fibers, and by 3.0, 2.8, 2.3, 3.6, 1.9, and 3.0 times, respectively, across the fibers, compared to

non-densified wood. For the parameter $R_{ku}$, there is an increase in the values of roughness along and across the fibers by 1.4 and 2.1 times, respectively.

The values of the roughness parameters $R_a$, $R_q$, $R_z$, $R_p$, $R_v$, and $R_{sk}$ for birch wood after treatment TC-210 were reduced by 3.0, 2.9, 2.7, 3.2, 2.3, and 4.3 times, respectively, along the fibers, and by 2.6, 2.3, 1.9, 3.6, 1.4, and 6.1 times, respectively, across the wood fibers, compared to non-densified wood. For the parameter $R_{ku}$, there was an increase in the values of roughness along and across the fibers by 1.4 and 2.6 times, respectively. Similar results were found for Norway wood [32].

The difference between the percentage of the reduction in the values of the surface roughness densified at 180 °C and at 210 °C for alder wood was smaller than for birch wood. This can be explained by the fact that birch wood is harder and has a higher density than alder wood, and thus is harder to densify; its roughness is reduced to a lesser extent at a lower temperature of 180 °C than at a higher temperature of 210 °C. Alder wood is softer and has a lower density, and as a result it is easier to densify than birch wood, even at 180 °C. Therefore, the difference in the values of roughness was less pronounced in alder than in birch wood with the treatments TC-180 and TC-210. For birch wood, the values of the surface roughness parameters $R_a$, $R_q$, $R_z$, $R_p$, $R_v$, and $R_{sk}$ along and across the fibers after TC-210 were 17.4% and 8.7%, 18.7% and 7.7%, 20.7% and 6.2%, 8.7% and 21.7%, 27.0% and 1.4%, and 27.8% and 34.5%, respectively, which were lower in terms of the values of these parameters after TC-180. The values of $R_{ku}$ along the fibers were 10.5% lower, and across the fibers they were 12.7% higher for the surface after treatment TC-210, compared with the surface after treatment TC-180.

The results of some past studies have also suggested that Douglas fir veneer compressed at a temperature of 210 °C had lower average roughness ($R_a$) values, revealing a better surface quality than those compressed at a temperature of 180 °C [30].

Skewness is used to measure the symmetry of the profile about the mean line, while kurtosis describes the sharpness of the probability density of the profile. For both of the investigated wood species, sanding and thermal compression produced surfaces with higher concentrations of material near the top of the roughness profile, as indicated by the negative $R_{sk}$ values in both directions of measurement (Table 2). These values suggest a predominance of valleys. Moreover, for both surfaces, $R_{ku} > 3$, which indicated that the distribution curve had relatively many high peaks and low valleys. The results are in good agreement with previous findings related to the surface roughnesses of alder and birch veneers that have been thermally densified at different temperatures and pressures [22,24]. In contrast, other authors [50] observed that sanded surfaces presented positive $R_{sk}$ in both directions of measurement.

### 3.2. Roughness of Sanded and Thermally Densified Varnished Surfaces

ANOVA analysis showed that the wood species and the method of pre-treatment of the substrate surface had a negligible effect on the roughness parameters $R_q$, $R_z$, and $R_v$, but they significantly affected the parameters $R_a$, $R_p$, $R_{sk}$, and $R_{ku}$. The rest of the factors significantly affected the roughness parameters of the varnished surface (Table 3). When analyzing the impact of the investigated factors on the surface roughness changes, it is worth noting that there were considerable differences in the influence of these factors. The type of varnish and the direction of the wood fibers had the strongest influences on all of the roughness parameters. Among the studied factors, the effect of the pre-treatment of substrate surface on the values of the parameters $R_a$, $R_p$, and $R_{ku}$ was the weakest. The roughness values of the surfaces varnished with three different varnishes measured along the fibers were smaller than the values measured across the fibers. This was in good agreement with the results obtained by other authors [46]. They showed that compared to the parameters obtained along the fibers, the surface roughness parameters of coated ash wood across the fibers are higher by several fold. Since the direction of the wood fibers significantly affects all of the roughness parameters, and wood species significantly affects only $R_a$, $R_p$, $R_{sk}$, and $R_{ku}$, the analysis of the influence of the investigated factors

on the value of surface roughness depending on the direction of wood fibers is presented separately below for alder and birch wood.

**Table 3.** Analysis of variance of surface roughness for varnished samples.

| Source of Variation | F Value | | | | | | |
|---|---|---|---|---|---|---|---|
| | $R_a$ | $R_q$ | $R_z$ | $R_p$ | $R_v$ | $R_{sk}$ | $R_{ku}$ |
| Wood species (WS) | 7.006 * | 1.768 ** | 1.565 ** | 11.929 * | 0.050 ** | 4.299 * | 7.432 * |
| Direction (D) | 104.250 * | 83.661 * | 207.110 * | 46.082 * | 316.185 * | 107.371 * | 33.026 * |
| Varnish (V) | 218.427 * | 158.492 * | 337.711 * | 208.054 * | 366.002 * | 124.913 * | 32.558* |
| Layers (L) | 32.096 * | 25.554 * | 58.738 * | 46.382 * | 58.319 * | 6.437 * | 8.234 * |
| Pre-treatment (PT) | 3.375 * | 0.774 ** | 0.655 ** | 11.754 * | 0.414 ** | 8.453 * | 3.383 * |
| WS × D | 0.064 ** | 0.796 ** | 2.762 ** | 0.105 ** | 7.397 * | 9.981 * | 13.259 * |
| WS × V | 4.119 * | 0.871 ** | 0.682 ** | 4.421 * | 0.040 ** | 1.611 ** | 3.014 ** |
| WS × L | 2.240 ** | 0.759 ** | 0.880 ** | 1.545 ** | 0.760 ** | 1.193 ** | 1.058 ** |
| WS × PT | 8.501 * | 7.984 * | 16.828 * | 7.147 * | 21.958 * | 4.919 * | 3.227 * |
| D × V | 10.174 * | 10.838 * | 35.941 * | 4.865 * | 63.837 * | 32.059 * | 15.119 * |
| D × L | 1.719** | 2.007** | 4.446* | 0.987** | 7.669* | 0.488** | 1.406** |
| D × PT | 3.620 * | 2.737 ** | 4.191 * | 4.475 * | 2.779 ** | 2.241 ** | 2.046 ** |
| V × L | 10.525 * | 7.838 * | 17.456 * | 15.970 * | 16.092 * | 2.143 * | 1.888 ** |
| V × PT | 32.741 * | 21.854 * | 54.443 * | 62.354 * | 37.724 * | 2.996 * | 1.249 ** |
| L × PT | 0.850 ** | 0.997 ** | 1.046 ** | 0.915 ** | 1.235 ** | 0.967 ** | 0.363 ** |
| WS × D × V | 1.277 ** | 0.299 ** | 0.208 ** | 2.557 ** | 1.343 ** | 9.343 * | 1.069 ** |
| WS × D × L | 0.932 ** | 0.912 ** | 1.099 ** | 0.687 ** | 1.590 ** | 2.043 ** | 0.989 ** |
| WS × D × PT | 0.231 ** | 0.828 ** | 1.878 ** | 0.165 ** | 3.982 * | 0.469 ** | 1.046 ** |
| WS × V × L | 2.158 * | 1.573 ** | 1.831 ** | 1.371 ** | 2.127 * | 1.280 ** | 0.464 ** |
| WS × V × PT | 5.833 * | 3.998 * | 6.730 * | 8.080 * | 5.389 * | 2.305 ** | 1.017 ** |
| WS × L × PT | 5.553 * | 3.975 * | 5.198 * | 5.626 * | 5.429 * | 1.289 ** | 0.105 ** |
| D × V × L | 0.872 ** | 0.687 ** | 1.076 ** | 0.432 ** | 1.986 ** | 0.977 ** | 1.575 ** |
| D × V × PT | 0.728 ** | 0.944 ** | 3.040 * | 1.740 ** | 2.944 * | 1.188 ** | 2.236 ** |
| V × L × PT | 0.145 ** | 0.023 ** | 0.383 ** | 0.241 ** | 0.391 ** | 1.218 ** | 0.070 ** |
| V × L × PT | 1.245 ** | 0.563 ** | 1.020 ** | 0.973 ** | 1.905* * | 1.818 ** | 2.088 * |
| WS × D × V × L | 0.546 ** | 0.389 ** | 0.431 ** | 0.416 ** | 0.289 ** | 0.855 ** | 0.577 ** |
| WS × D × V × PT | 1.534 ** | 1.500 ** | 2.671 * | 0.639 ** | 3.805 * | 2.753 * | 1.679 ** |
| WS × D × L × PT | 0.408 ** | 0.383 ** | 0.467 ** | 0.046 ** | 1.461 ** | 1.277 ** | 0.802 ** |
| WS × V × L × PT | 3.476 * | 2.203 * | 3.038 * | 3.090 * | 2.768 * | 1.222 ** | 0.865 ** |
| D × V × L × PT | 0.782 ** | 0.489 ** | 0.962 ** | 0.550 ** | 1.124 ** | 0.373 ** | 0.295 ** |
| WS × D × V × L × PT | 0.356 ** | 0.496 ** | 0.226 ** | 0.335 ** | 0.472 ** | 2.203 * | 0.468 ** |

*: Significant ($p \leq 0.05$); **: Non-significant.

### 3.2.1. Surface Roughness along the Fibers for Varnished Alder Wood Samples

According to the ANOVA analysis, the parameters $R_a(\parallel)$, $R_q(\parallel)$, $R_z(\parallel)$, $R_p(\parallel)$, $R_v(\parallel)$, and $R_{sk}(\parallel)$ were most strongly influenced by the type of varnish, the number of varnish layers (except $R_{sk}(\parallel)$, which was affected insignificantly) had a weaker effect, and the method of pre-treatment of the substrate surface (except $R_{sk}(\parallel)$, which was affected insignificantly) had the weakest effect. The type of varnish, the number of varnish layers, and the method of pre-treatment of the substrate surface had a negligible effect on the parameter $R_{ku}(\parallel)$.

The lowest values of the roughness parameters $R_a(\parallel)$, $R_q(\parallel)$, $R_z(\parallel)$, $R_p(\parallel)$, $R_v(\parallel)$, and $R_{ku}(\parallel)$ were recorded for the UV-varnished surface (1.30 µm, 1.63 µm, 6.69 µm, 3.55 µm, 3.22 µm, and 3.28 µm, respectively), while the lowest $R_{sk}(\parallel) = -0.66$ µm—for the PUR-varnished surface. The highest values of the roughness parameters $R_a(\parallel)$, $R_q(\parallel)$, $R_z(\parallel)$, $R_p(\parallel)$, and $R_v(\parallel)$ were recorded for the WB-varnished surface (5.44 µm, 6.91 µm, 33.19 µm, 14.42 µm, and 18.77 µm, respectively). The highest values of the roughness parameters $R_{sk}(\parallel)$ and $R_{ku}(\parallel)$ were observed for UV-varnished ($R_{sk}(\parallel) = 0.09$ µm) and PUR-varnished ($R_{ku}(\parallel) = 3.96$ µm) surfaces, respectively. The values of the parameters $R_a(\parallel)$, $R_q(\parallel)$, $R_z(\parallel)$, $R_p(\parallel)$, and $R_v(\parallel)$ for the UV-varnished surface were, respectively, 3.2 and 4.2, 3.3 and 5.5,

3.9 and 5.0, 3.0 and 4.1, and 4.9 and 5.8 times lower than the values of these parameters for the PUR-varnished and WB-varnished surfaces (Figure 1). The values of $R_{sk}(\parallel)$ for the PUR-varnished surface were 1.5 and 7.0 times lower than for the WB-varnished and UV-varnished surfaces, respectively. The values of $R_{ku}(\parallel)$ for the UV-varnished surface were 1.1 and 1.2 times lower than for the WB-varnished and PUR-varnished surfaces, respectively.

The lowest values of the roughness parameters $R_a(\parallel)$, $R_q(\parallel)$, $R_z(\parallel)$, $R_p(\parallel)$, and $R_v(\parallel)$ were recorded for the surface varnished with two varnish layers with intermediate sanding (2S), and with three varnish layers (2.27 μm and 2.63 μm, 2.90 μm and 3.40 μm, 13.80 μm and 14.20 μm, 6.14 μm and 6.73 μm, and 7.66 μm and 7.47 μm, respectively) (Figure 1). In addition, the differences between these values were insignificant. The smallest values of $R_{sk}(\parallel)$ = −0.45 μm and $R_{sk}(\parallel)$ = −0.27 μm were recorded for surfaces with one varnish layer and two varnish layers with intermediate sanding (2S), respectively; the difference between these values was insignificant. The lowest values of $R_{ku}(\parallel)$ = 3.12 μm and $R_{ku}(\parallel)$ = 3.23 μm were recorded for surfaces with three varnish layers and two varnish layers without intermediate sanding (2NS), respectively; the difference between these values was insignificant. The highest values of the roughness parameters $R_a(\parallel)$, $R_q(\parallel)$, $R_z(\parallel)$, $R_p(\parallel)$, and $R_v(\parallel)$ were measured for the surface varnished with two varnish layers without intermediate grinding (2NS) and one varnish layer (3.57 μm and 4.50 μm, 4.51 μm and 5.75 μm, 21.29 μm and 28.14 μm, 9.36 μm and 11.86 μm, and 11.91 μm and 16.35 μm, respectively). The highest values of the roughness parameters $R_{sk}(\parallel)$ were observed for surfaces that had been varnished with two varnish layers without intermediate sanding (2NS) and with three varnish layers ($R_{sk}(\parallel)$ = −0.22 μm and $R_{sk}(\parallel)$ = −0.19 μm, respectively); the difference between them was insignificant. The highest values of the roughness parameters $R_{ku}(\parallel)$ were observed for surfaces with two varnish layers with intermediate sanding (2S) and with one varnish layer ($R_{ku}(\parallel)$ = 3.59 μm and $R_{ku}(\parallel)$ = 3.88 μm, respectively); the difference between them was insignificant.

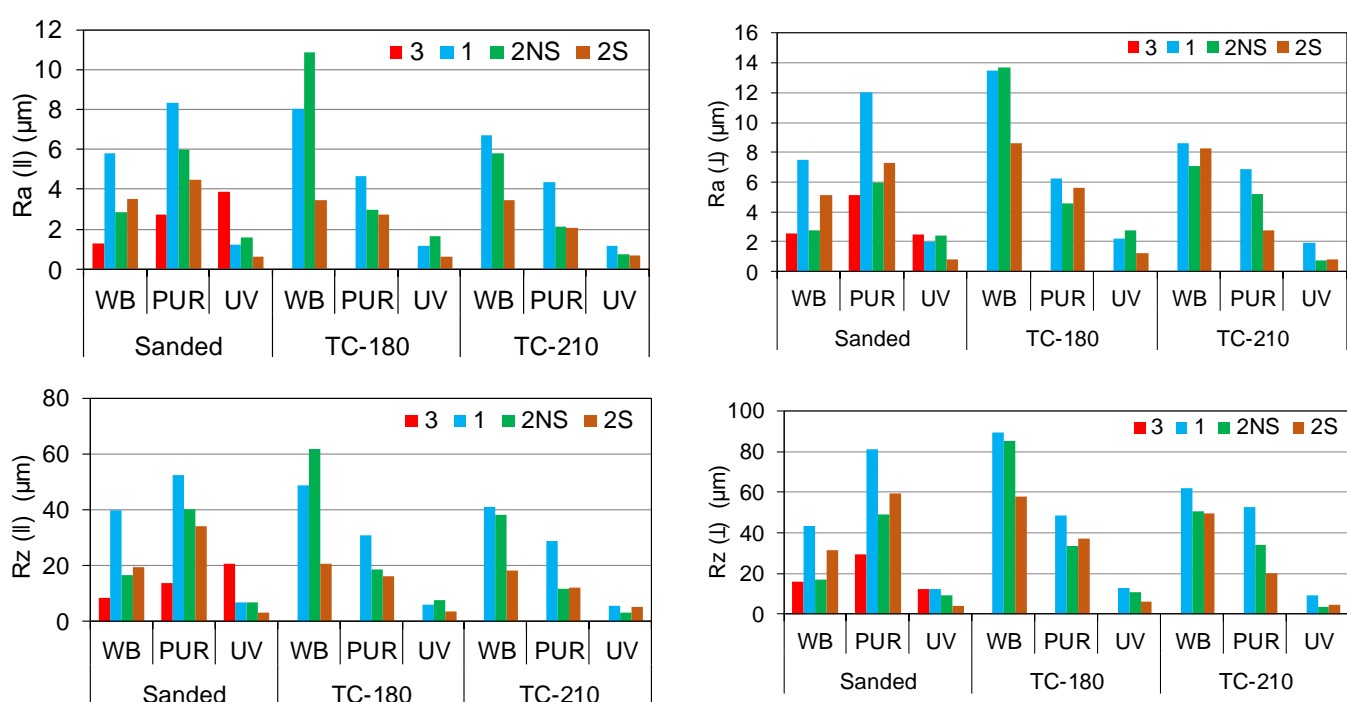

**Figure 1.** *Cont.*

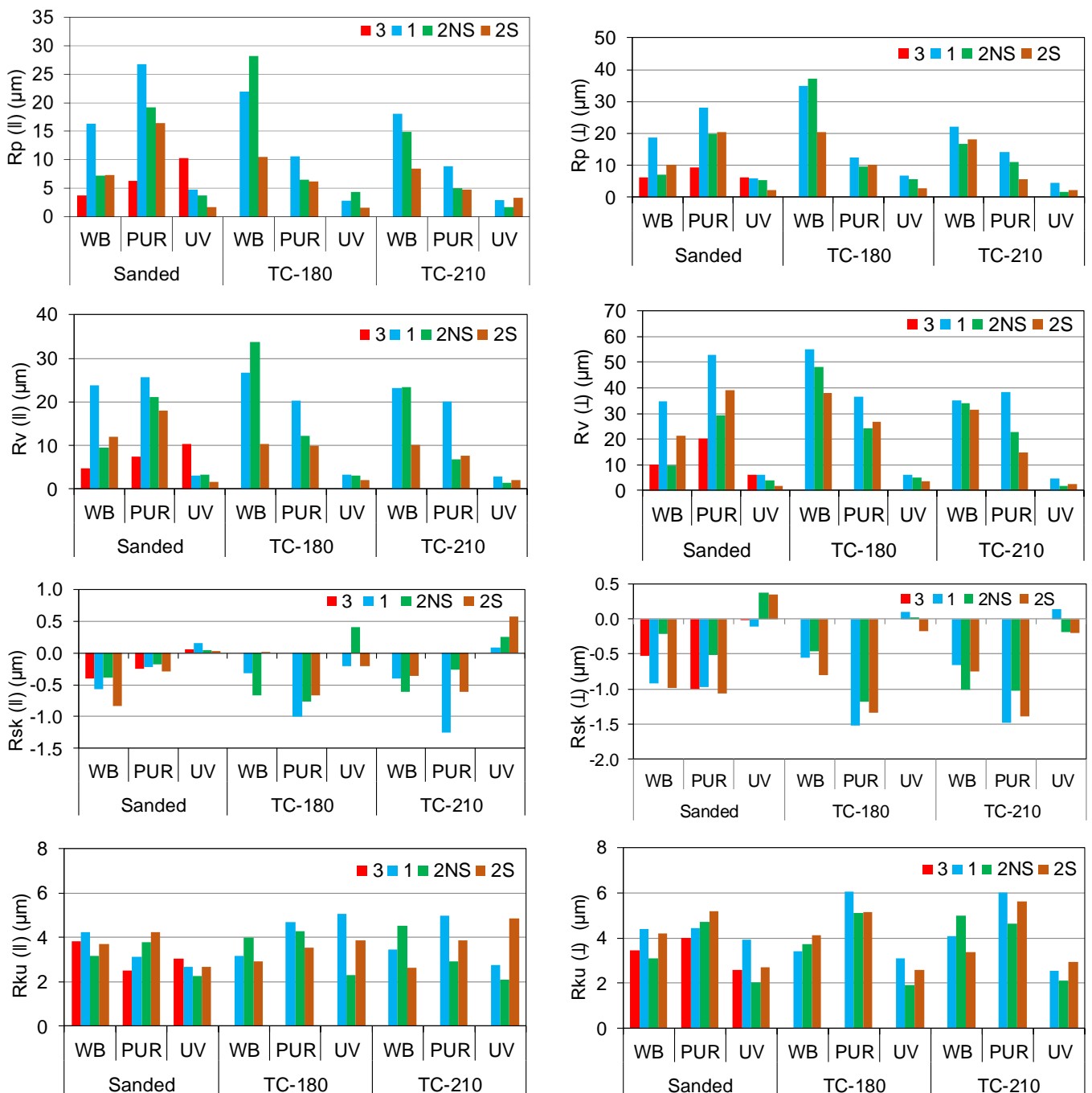

**Figure 1.** Surface roughness of alder wood samples along and across the fibers.

The lowest values of the roughness parameters $R_a(\|)$, $R_q(\|)$, $R_z(\|)$, $R_p(\|)$, and $R_v(\|)$ were recorded for the surfaces after TC-210 treatment and sanding (S), and the highest value was observed for the surface after TC-180 treatment (Figure 1). The values of the roughness parameters $R_a(\|)$, $R_q(\|)$, $R_z(\|)$, $R_p(\|)$, and $R_v(\|)$ for the surfaces after TC-210 treatment and sanding were 3.27 μm and 3.49 μm, 4.19 μm and 4.47 μm, 20.04 μm and 21.53 μm, 8.11 μm and 10.11 μm, and 11.94 μm and 11.51 μm, respectively. The values of the roughness parameters $R_a(\|)$, $R_q(\|)$, $R_z(\|)$, $R_p(\|)$, and $R_v(\|)$ for the surface after TC-180 treatment were 3.99 μm, 5.06 μm, 24.04 μm, 10.15 μm, and 13.87 μm, respectively. The difference between the values of the roughness parameters for these surfaces was insignificant. The values of $R_{sk}(\|)$ after the treatments TC-180 and TC-210 were −0.42 μm and −0.34 μm, respectively, and these were lower than after sanding ($R_{sk}(\|)$ = −0.23 μm). The difference between the

values of the roughness parameter $R_{sk}(\parallel)$ for these surfaces was insignificant. The difference between the values of the roughness parameter $R_{ku}(\parallel)$ for surfaces after the treatments TC-180 and TC-210 was also insignificant.

3.2.2. Surface Roughness across the Fibers for Varnished Alder Wood Samples

According to the ANOVA analysis, the parameters $R_a(\perp)$, $R_q(\perp)$, $R_z(\perp)$, $R_p(\perp)$, $R_v(\perp)$, and $R_{ku}(\perp)$ were most strongly influenced by the type of varnish, and less affected by the number of varnish layers (except $R_{ku}(\perp)$, which was affected insignificantly) and least affected by the method of pre-treatment of the substrate surface (except $R_{ku}(\perp)$, which was affected insignificantly). The parameter $R_{sk}(\perp)$ was most strongly influenced by the type of varnish, less affected by the method of pre-treatment of the substrate surface, and was least affected by the number of varnish layers.

The lowest values of the roughness parameters $R_a(\perp)$, $R_q(\perp)$, $R_z(\perp)$, $R_p(\perp)$, $R_v(\perp)$, and $R_{ku}(\perp)$ were recorded for the UV-varnished surface (1.79 μm, 2.17 μm, 8.75 μm, 4.48 μm, 4.27 μm, and 2.68 μm, respectively) (Figure 1), while the lowest $R_{sk}(\perp) = -1.21$ μm for the PUR-varnished surface. The highest values of the roughness parameters $R_a(\perp)$, $R_q(\perp)$, $R_z(\perp)$, $R_p(\perp)$, and $R_v(\perp)$ were measured for the WB-varnished surface (8.35 μm, 10.71 μm, 54.68 μm, 20.72 μm, and 33.96 μm, respectively). The highest values of the roughness parameters $R_{sk}(\perp)$ and $R_{ku}(\perp)$ were observed for the UV-varnished ($R_{sk}(\perp) = 0.05$ μm) and PUR-varnished ($R_{ku}(\perp) = 5.26$ μm) surfaces, respectively. The values of the parameters $R_a(\perp)$, $R_q(\perp)$, $R_z(\perp)$, $R_p(\perp)$, and $R_v(\perp)$ for the UV-varnished surface were 3.5 and 4.7, 3.8 and 4.9, 5.2 and 6.2, 3.1 and 4.6, and 7.4 and 8.0 times smaller than the values of these parameters for the PUR-varnished and WB-varnished surfaces, respectively. The values of $R_{sk}(\perp)$ for the PUR-varnished surface were 1.8 and 26.0 times smaller than for the WB-varnished and UV-varnished surfaces, respectively. The values of $R_{ku}(\perp)$ for the UV-varnished surface were 1.4 and 2.0 times lower than for the WB-varnished and PUR-varnished surfaces, respectively.

The lowest values of the roughness parameters $R_a(\perp)$, $R_q(\perp)$, $R_z(\perp)$, $R_p(\perp)$, and $R_v(\perp)$ were fixed for the surface varnished with three varnish layers and two varnish layers with intermediate sanding (2S) (3.38 μm and 4.39 μm, 4.35 μm and 5.72 μm, 19.16 μm and 28.90 μm, 7.18 μm and 9.78 μm, and 11.97 μm and 19.12 μm, respectively) (Figure 1); in addition, the difference between these values for the parameters $R_a(\perp)$, $R_q(\perp)$, and $R_p(\perp)$ was insignificant. The surfaces 2S and 2NS differed insignificantly in terms of $R_a(\perp)$, $R_q(\perp)$, $R_z(\perp)$, and $R_v(\perp)$, whereas the smallest values of $R_{sk}(\perp) = -0.69$ μm and $R_{sk}(\perp) = -0.66$ μm were recorded for surfaces with two varnish layers with intermediate sanding (2S) and one varnish layer, respectively; the difference between these values was insignificant. The lowest values of $R_{ku}(\perp) = 3.36$ μm and $R_{ku}(\perp) = 3.59$ μm were measured for surfaces with three varnish layers and two varnish layers without intermediate sanding (2NS), respectively; the difference between these values was insignificant. The highest values of the roughness parameters $R_a(\perp)$, $R_q(\perp)$, $R_z(\perp)$, $R_p(\perp)$, and $R_v(\perp)$ were measured for the surfaces varnished with two varnish layers without intermediate sanding (2NS), and with one varnish layer (5.02 μm and 6.68 μm, 6.42 μm and 8.66 μm, 32.53 μm and 45.63 μm, 12.66 μm and 16.08 μm, and 19.87 μm and 29.56 μm, respectively). The highest values of the roughness parameter $R_{sk}(\perp)$ were observed for surfaces with three varnish layers and two varnish layers without intermediate sanding (2NS) ($R_{sk}(\perp) = -0.51$ μm and $R_{sk}(\perp) = -0.46$ μm, respectively); the difference between them was insignificant). The highest values of the roughness parameters $R_{ku}(\perp)$ were observed for surfaces with two varnish layers with intermediate sanding (2S) and for one varnish layer ($R_{ku}(\perp) = 3.95$ μm and $R_{ku}(\perp) = 4.22$ μm, respectively); the difference between them was insignificant.

The lowest values of the roughness parameters $R_a(\perp)$, $R_q(\perp)$, $R_z(\perp)$, $R_p(\perp)$, and $R_v(\perp)$ were recorded for surfaces after sanding and treatment TC-210, and the highest values for the surface after treatment TC-180 (Figure 1). The values of the roughness parameters $R_a(\perp)$, $R_q(\perp)$, $R_z(\perp)$, $R_p(\perp)$, and $R_v(\perp)$ for surfaces after sanding and treatment TC-210 were 4.60 μm and 4.92 μm, 5.95 μm and 6.40 μm, 29.48 μm and 33.65 μm, 11.94 μm and

11.19 μm, and 19.00 μm and 21.61 μm, respectively. The difference between the values of the roughness parameters for these surfaces was insignificant. The values of the roughness parameters $R_a(\perp)$, $R_q(\perp)$, $R_z(\perp)$, $R_p(\perp)$, and $R_v(\perp)$ for the surface after the TC-180 treatment were 6.70 μm, 8.62 μm, 44.45 μm, 16.10 μm, and 28.36 μm, respectively. The values of $R_{sk}(\perp)$ after the treatments TC-210 and TC-180 ($R_{sk}(\perp) = -0.71$ μm and $R_{sk}(\perp) = -0.66$ μm, respectively) were lower than those values after sanding ($R_{sk}(\perp) = -0.45$ μm). The difference between the values of the roughness parameter $R_{sk}(\perp)$ for these surfaces after the treatments TC-210 and TC-180 was insignificant.

3.2.3. Surface Roughness along the Fibers for Varnished Birch Wood Samples

According to the ANOVA analysis, the parameters $R_a(\parallel)$, $R_q(\parallel)$, $R_z(\parallel)$, $R_p(\parallel)$, $R_v(\parallel)$, and $R_{sk}(\parallel)$ are most strongly influenced by the type of varnish (except $R_{sk}(\parallel)$, which was insignificantly affected), were less affected by the number of varnish layers, and were least affected by the method of pre-treatment of the substrate surface (except for $R_a(\parallel)$, $R_q(\parallel)$, and $R_{sk}(\parallel)$, which were affected insignificantly). The type of varnish, the number of varnish layers, and the method of treatment of the substrate surface had a negligible effect on the parameter $R_{ku}(\parallel)$. From a practical point of view, it is important that there was no difference between the sanded and thermally densified surfaces, in terms of surface roughness, at 210 °C and at 180 °C.

The lowest values of the roughness parameters $R_a(\parallel)$, $R_q(\parallel)$, $R_z(\parallel)$, $R_p(\parallel)$, $R_v(\parallel)$, and $R_{ku}(\parallel)$ were recorded for the UV-varnished surface (0.96 μm, 1.23 μm, 5.67 (m, 2.51 μm, 3.15 μm, and 2.89 μm, respectively), while the lowest $R_{sk}(\parallel) = -0.35$ μm for the PUR-varnished surface (Figure 2). The highest values of the roughness parameters $R_a(\parallel)$, $R_q(\parallel)$, $R_z(\parallel)$, $R_p(\parallel)$, $R_v(\parallel)$, and $R_{ku}(\parallel)$ were measured for the WB-varnished surface (4.50 μm, 5.85 μm, 27.78 μm, 12.39 μm, 15.39 μm, and 3.77 μm, respectively). The highest value of the roughness parameter $R_{sk}(\parallel)$ was observed for the UV-varnished surface ($R_{sk}(\parallel) = -0.12$ μm). The values of the parameters $R_a(\parallel)$, $R_q(\parallel)$, $R_z(\parallel)$, $R_p(\parallel)$, $R_v(\parallel)$, and $R_{ku}(\parallel)$ for the UV-varnished surface were 4.1 and 4.7, 4.1 and 4.8, 4.3 and 4.9, 4.0 and 4.9, 4.5 and 4.9, and 1.3 and 1.3 times smaller than the values of these parameters for the PUR-varnished and the WB-varnished surfaces, respectively. The UV-varnished, PUR-varnished, and WB-varnished surfaces differed insignificantly for the parameter $R_{sk}(\parallel)$. The values of $R_{sk}(\parallel)$ for the PUR-varnished surface were 1.4 and 3.0 times smaller than for the WB-varnished and UV-varnished surfaces, respectively.

The lowest values of the roughness parameters $R_a(\parallel)$, $R_q(\parallel)$, $R_z(\parallel)$, $R_p(\parallel)$, $R_v(\parallel)$, and $R_{sk}(\parallel)$ were fixed for the surfaces varnished with two varnish layers with intermediate sanding (2S), and with three varnish layers (2.42 μm and 2.64 μm, 3.15 μm and 3.18 μm, 15.40 μm and 13.01 μm, 6.04 μm and 6.20 μm, 9.34 μm and 6.81 μm, and −0.43 μm and −0.25 μm, respectively) (Figure 2); in addition, the differences between these values were insignificant. The largest values of the roughness parameters $R_a(\parallel)$, $R_q(\parallel)$, $R_z(\parallel)$, $R_p(\parallel)$, $R_v(\parallel)$, and $R_{sk}(\parallel)$ were measured for the surfaces varnished with two varnish layers without intermediate sanding (2NS), and with one varnish layer (3.07 μm and 3.66 μm, 3.92 μm and 4.82 μm, 18.22 μm and 23.35 μm, 8.51 μm and 10.09 μm, 9.70 μm and 13.26 μm, and −0.07 μm and −0.22 μm, respectively); in addition, the difference between these values was insignificant (except for $R_v(\parallel)$). The varnished surfaces with three, one, 2NS, and 2S varnished layers differed insignificantly for the parameter $R_{ku}(\parallel)$.

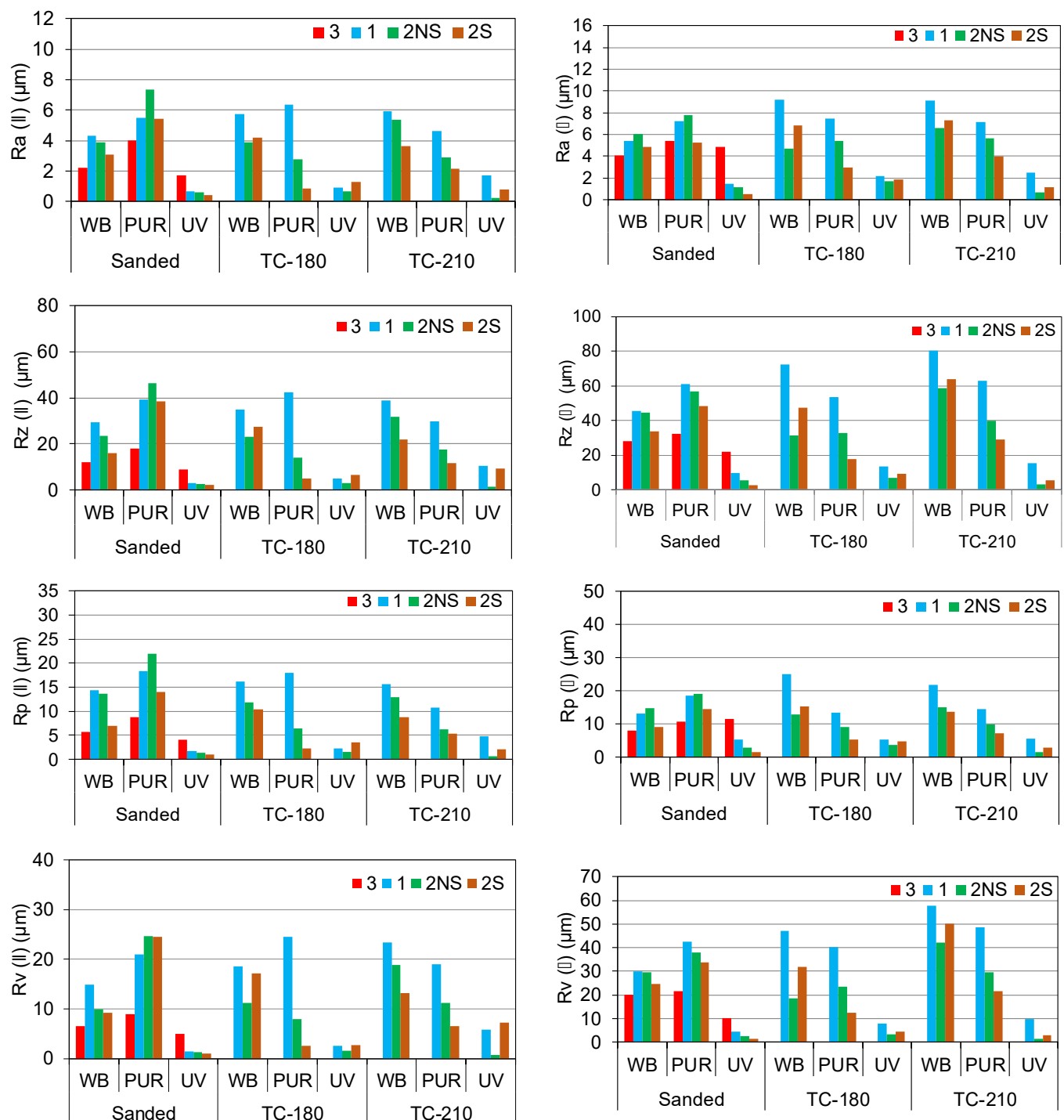

**Figure 2.** *Cont.*

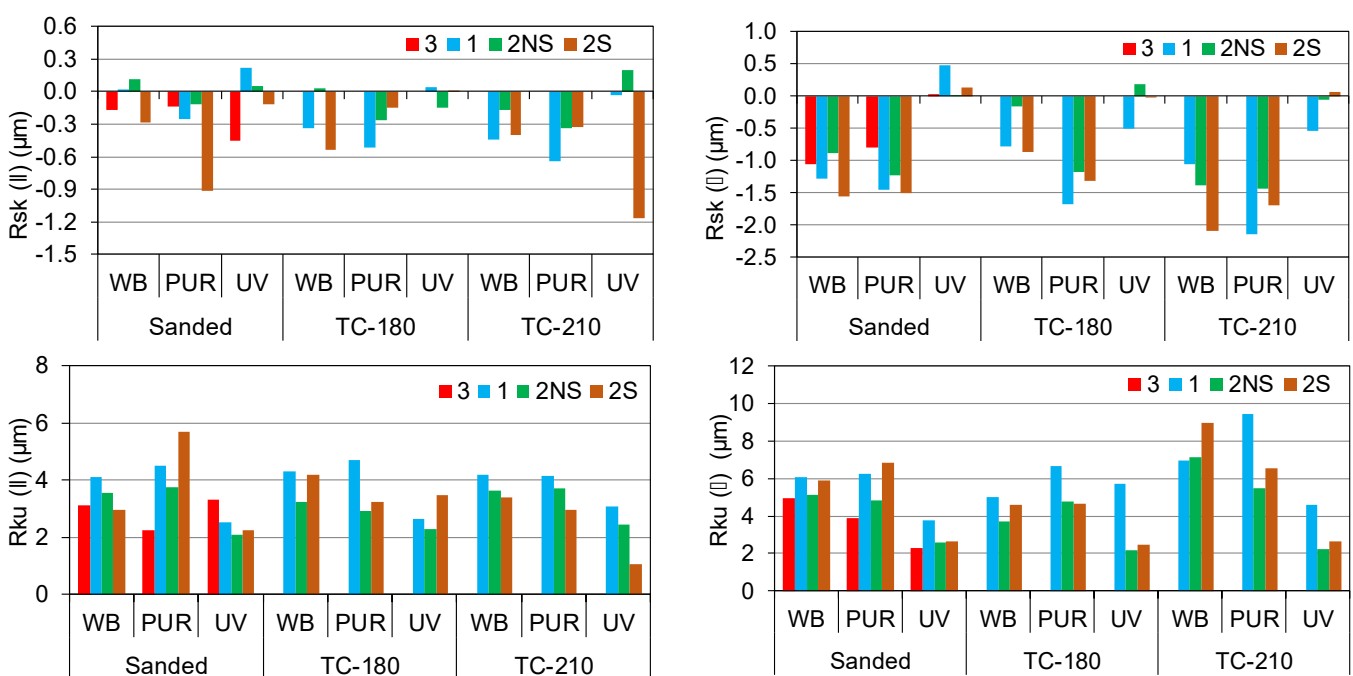

**Figure 2.** Surface roughnesses of alder wood samples along and across the fibers.

3.2.4. Surface Roughness across the Fibers for Varnished Birch Wood Samples

According to the ANOVA analysis, the parameters $R_a(\perp)$, $R_q(\perp)$, $R_z(\perp)$, $R_p(\perp)$, $R_v(\perp)$, $R_{sk}(\perp)$, and $R_{ku}(\perp)$ were most strongly influenced by the type of varnish, were less affected by the number of varnish layers, and were least affected by the method of pre-treatment of the substrate surface (except for $R_a(\perp)$, $R_q(\perp)$, and $R_p(\perp)$, which were affected insignificantly).

The lowest values of the roughness parameters $R_a(\perp)$, $R_q(\perp)$, $R_z(\perp)$, $R_p(\perp)$, $R_v(\perp)$, and $R_{ku}(\perp)$ were recorded for the UV-varnished surface (1.88 μm, 2.33 μm, 10.10 μm, 4.64 μm, 5.46 μm, and 3.45 μm, respectively), while the lowest $R_{sk}(\perp) = -1.52$ μm for the PUR-varnished surface (Figure 2). The highest values of the roughness parameters $R_a(\perp)$, $R_q(\perp)$, $R_z(\perp)$, $R_p(\perp)$, and $R_v(\perp)$ were fixed for the WB-varnished surface (6.88 μm, 10.28 μm, 54.85 μm, and 16.34 μm and 38.12 μm, respectively). The highest values of the roughness parameters $R_{sk}(\perp)$ and $R_{ku}(\perp)$ were observed for the UV-varnished ($R_{sk}(\perp) = -0.10$ μm) and PUR-varnished ($R_{ku}(\perp) = 6.30$ μm) surfaces, respectively. The values of the parameters $R_a(\perp)$, $R_q(\perp)$, $R_z(\perp)$, $R_p(\perp)$, and $R_v(\perp)$ for the UV-varnished surface were 3.2 and 3.6, 3.5 and 4.4, 4.5 and 5.4, 2.7, and 3.5, and 6.1 and 7.0 times lower than the values of these parameters for the PUR-varnished and WB-varnished surfaces, respectively. The values of $R_{sk}(\perp)$ for the PUR-varnished surface were 1.4 and 15.1 times smaller than for the WB-varnished and UV-varnished surfaces, respectively. The values of $R_{ku}(\perp)$ for the UV-varnished surface were 1.7 and 1.8 times smaller than for the WB-varnished and PUR-varnished surfaces, respectively.

The lowest values of the roughness parameters $R_a(\perp)$, $R_q(\perp)$, and $R_p(\perp)$ were recorded for surfaces that had been varnished with two varnish layers with intermediate (2S) and without intermediate (2NS) sanding (3.84 μm and 4.41 μm, 5.16 μm and 5.85 μm, and 8.27 μm and 9.89 μm, respectively) (Figure 2); in addition, the differences between these values for the parameters $R_a(\perp)$, $R_q(\perp)$, and $R_p(\perp)$ was insignificant. The surfaces with 2NS and three varnish layers did not differ in terms of $R_a(\perp)$, $R_q(\perp)$, and $R_p(\perp)$. The lowest values of $R_z(\perp)$ and $R_v(\perp)$ were fixed for surfaces with three varnish layers and with two varnish layers with intermediate sanding (2S) ($R_z(\perp) = 27.38$ μm and $R_z(\perp) = 28.56$ μm, and $R_v(\perp) = 17.28$ μm and $R_v(\perp) = 20.29$ μm, respectively); in addition, they differed insignificantly. The largest values of $R_z(\perp) = 47.46$ μm and $R_v(\perp) = 33.35$ μm were observed for the surface with one varnish layer. Surfaces with 2S and 2NS varnish layers differed insignificantly in terms of $R_z(\perp)$ and $R_v(\perp)$, whereas, the smallest values of $R_{sk}(\perp) = -1.04$ μm and

$R_{sk}(\perp) = -0.98$ µm were measured for surfaces with one varnish layer and two varnish layers with intermediate sanding (2S), respectively; the differences between these values were insignificant. The lowest values of $R_{ku}(\perp) = 3.71$ µm and $R_{ku}(\perp) = 4.23$ µm were evaluated for surfaces with three varnish layers and two varnish layers without intermediate sanding (2NS), respectively; the difference between these values was insignificant. The highest values of the roughness parameter $R_{sk}(\perp)$ were observed for surfaces with two varnish layers without intermediate sanding (2NS) and with three varnish layers ($R_{sk}(\perp) = -0.68$ µm and $R_{sk}(\perp) = -0.61$ µm, respectively); the difference between them was insignificant. The highest values of the roughness parameters $R_{ku}(\perp)$ were observed for surfaces with two varnish layers with intermediate sanding (2S) and with one varnish layer ($R_{ku}(\perp) = 5.04$ µm and $R_{ku}(\perp) = 6.19$ µm, respectively); the difference between them was insignificant.

From a practical point of view, it is important that there was no difference in terms of surface roughness between the sanded surface and surfaces that had been thermally densified at temperatures of 180 °C and 210 °C. The surfaces after sanding and thermal compression at both temperatures differed insignificantly for the parameters $R_a(\perp)$, $R_q(\perp)$, and $R_p(\perp)$. The surfaces after sanding and treatment with TC-180 differed insignificantly for the parameters $R_z(\perp)$, $R_v(\perp)$, $R_{sk}(\perp)$, and $R_{ku}(\perp)$.

### 3.3. Effect of the Type of Varnish and the Number of Varnish Layers on the Surface Roughness of Samples

For both of the investigated wood species, the parameters $R_a$, $R_q$, $R_z$, $R_p$, and $R_v$ along and across the fibers were most strongly influenced by the type of varnish, were less affected by the number of varnish layers, and were least affected by the method of pre-treatment of the substrate surface. The lowest values of the roughness parameters were recorded for the UV-varnished surface, and then for WB-varnished and the PUR-varnished surfaces (Figures 1 and 2). The values of the parameters $R_a$, $R_q$, $R_z$, $R_p$, and $R_v$ for the UV-varnished surfaces were, respectively, 3.0–5.8 and 3.1–8.0 times smaller along and across the fibers of alder wood, and 4.0–4.9 and 2.7–7.0 times smaller along and across the fibers of birch wood than the values of these parameters for the PUR-varnished and WB-varnished surfaces.

The values of skewness $R_{sk}$ for alder wood along and across the fibers for the UV-varnished surface were positive (+), and for the WB-varnished and PUR-varnished surfaces, they were negative (−). This means that the UV-varnished surface is characterized by a profile with the valleys filled in or by high peaks, while the WB-varnished and PUR-varnished surfaces are characterized by profiles with the peaks removed, or by deep scratches. In addition, the UV-varnished, WB-varnished and PUR-varnished surfaces along the fibers and WB-varnished and PUR-varnished surfaces across the fibers had kurtosis $R_{ku} > 3$ (relatively many high peaks and low valleys), and the UV-varnished surface across the fibers had kurtosis $R_{ku} < 3$ (relatively few high peaks and low valleys).

Another surface profile was observed for birch wood. The values of skewness $R_{sk}$ for birch wood along and across the fibers for surfaces that were varnished with all types of varnishes were negative (−). This indicates that the bulk of the material of the sample was above the mean line. The values of kurtosis $R_{ku}$ for the WB-varnished and PUR-varnished surfaces along the fibers, and for the UV-varnished, WB-varnished, and PUR-varnished surfaces across the fibers were $R_{ku} > 3$ (relatively many high peaks and low valleys), while for the UV-varnished surface along the fibers it was $R_{ku} < 3$ (relatively few high peaks and low valleys).

The number of varnish layers significantly affects the surface roughness. The lowest values of surface roughness for both wood species in both directions of measurement were observed for surfaces with two varnish layers with intermediate sanding (2S) and three varnish layers. In contrast, the highest degree of roughness was observed for surfaces with two varnish layers without intermediate sanding (2NS) and with one varnish layer. Surfaces with all of the investigated numbers of varnish layers were characterized by a profile with

a negative (−) value of skewness $R_{sk}$ (the bulk of the material of the sample was above the mean line) and kurtosis $R_{ku} > 3$ (relatively many high peaks and low valleys). Thus, two varnish layers with intermediate sanding (2S) can be recommended for application in practice. An application of three varnish layers is unprofitable for economic reasons and is also technologically time-consuming. An application of one varnish layer is cost-effective, but it does not provide a required surface roughness. It was observed that the sanded samples with three product coatings presented smoother and better finished surfaces, although two coats of the finishing product were sufficient to reduce the roughness [51].

From a practical point of view, it is important that there was no difference in terms of surface roughness between the surfaces that had been sanded or thermally densified at a temperature of 210 °C. This confirmed our assumption that the sanding operation as a pre-treatment of the wood surface before varnishing can be replaced by the operation of thermal compression.

## 4. Conclusions

The feasibility of reducing the need for sanding is considered. In this study, no significant difference was found between the values of the surface roughness for alder and birch wood veneers. For sanded and thermally densified alder and birch veneer surfaces, the roughness values measured across the fibers were several times greater than those measured along the fibers. The surfaces of the alder and birch veneers after sanding and thermal densification TC-210 at a temperature of 210 °C differed insignificantly ($p > 0.05$) from each other in terms of the roughness parameters $R_a$, $R_q$, and $R_p$. The obtained results confirmed our assumption that the time-consuming operation of pre-treatment of the wood surface before finishing by sanding can be replaced by the operation of thermal compression. This replacement, in addition to providing the necessary surface roughness for finishing, makes this surface more attractive in terms of decorative properties. Therefore, in terms of surface roughness and decorative properties, the recommended temperature for thermal compression could be 210 °C.

The wood species and the method of pre-treatment of the substrate surface had a negligible effect on the roughness parameters $R_q$, $R_z$, and $R_v$, but they significantly affected the parameters $R_a$, $R_p$, $R_{sk}$, and $R_{ku}$ of the varnished surface. The type of varnish and the number of varnish layers significantly affected all of the studied surface roughness parameters. The UV-varnished and WB-varnished surfaces were characterized by the lowest and highest degrees of surface roughness, respectively. The highest values of the roughness parameters were observed for surfaces with one and two varnish layers without intermediate sanding (2NS), while the lowest degrees of roughness were shown by surfaces with three varnish layers and with two varnish layers with intermediate sanding (2S). The roughness decreased with the number of layers in the coating.

Therefore, in terms of the surface roughness and the decorative properties of the finish, a temperature of 210 °C for the thermal compression of the wood surface, then finishing with a UV varnish with two varnish layers with intermediate sanding (2S) can be recommended as an application in practice.

**Author Contributions:** Conceptualization, P.B.; methodology, P.B., T.K., B.L. and N.B.; investigation, T.K., B.L. and N.B.; writing—original draft preparation, P.B.; writing—review and editing, P.B., T.K., B.L. and N.B.; project administration, P.B. All authors have read and agreed to the published version of the manuscript.

**Funding:** This research was supported by the Polish National Agency for Academic Exchange (NAWA) under contract No. PPN/ULM/2020/1/00188/U/00001, to implement the project "Development of a Novel Wood Surface Preparation Method before Varnishing" by Prof. Pavlo Bekhta at the Poznan University of Life Science, Poland.

**Institutional Review Board Statement:** Not applicable.

**Informed Consent Statement:** Not applicable.

**Data Availability Statement:** The data that support the findings of this study are available upon reasonable request from the authors.

**Acknowledgments:** Pavlo Bekhta acknowledges the Polish National Agency for Academic Exchange (NAWA) for the support of his research.

**Conflicts of Interest:** The authors declare no conflict of interest.

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
