# Peer review of "Surface Roughness of Varnished Wood Pre-Treated Using Sanding and Thermal Compression"

_forests, doi:10.3390/f13050777_

Round 1

Reviewer 1 Report

The authors have studied the potential impact of a new pre-treatment process of wood surface by its thermal compression on surface roughness. They was proposed to replace the sanding process of wood surface before its finishing with the thermal compression.

The manuscript is an original scientific work and well written. However, the text is too long and not very easy to follow. So, the manuscript can be shortened, if possible. The information given in introduction section is adequate. Methods were clear and findings were discussed sufficiently with the literature. 

On the other hand, some points indicated below need to be clarified:

- Rotary-peeled black alder and birch veneers were indicated to be used in the study. However, it is unclear whether the veneer sheets were produced by the authors or purchased ready-made.

- How about the veneer manufacturing conditions? in order to obtain a flat veneer with a flat surface and a uniform thickness during the rotary peeling process, it is necessary to ensure optimum rotary peeling conditions, such as angular parameters, peeling speed, and rotation. The position of the peeling knife, etc. This condition is determined by the type of wood, the diameter of the log, the thickness of the veneer, the heat treatment of the wood, and the precision of the peeling knife.

- Heating (cooking or steaming pre-treatment) of the birch and alder veneers were not mentioned in the manuscript. The heat pre-treatment of the veneer logs is an important stage affecting the veneer surface quality. So, log pre-treatment conditions before rotary cutting should be mentioned.  

Reviewer 2 Report

Materials and methods

An important part of this research is thermal compression and thus the method of thermal compression should be explained in more detail:

What was the duration of thermal Compression?

Were both press plates hot? What was the pressure?

Results

The results section is a bit large and confusing and should be revised, significantly. It is better to be more concise. Also, it is better to remove Figures 1, 2, 4 and 6.

Reviewer 3 Report

This manuscript reports the investigation on the surface roughness properties of varnished wood pre-treated by sanding and thermal compression. Major revision is requested. My comments are as below:

  1. The surface morphology images in micro-level should be provided.
  2. The chemical contents of the surface should be investigated after two different temperature thermal compressions.
  3. More detailed description of the three varnish systems should be given.
  4. In-deep discussion on the relationship between the surface property and the micro-level morphology should be given.
  5. As long as we know, sanding and compression treatments are traditional methods to improve the surface properties of wood products. The authors should enhance the background to distinguish their work from previously published reports.

Round 2

Reviewer 2 Report

  1. The results part of manuscript is still long and should be summarized as indicated before . Some figures should be removed!
  2. Another paper has already been published by the same authors in this journal entitled as” The Impact of Sanding and Thermal Compression of Wood, Varnish Type and Artificial Aging in Indoor Conditions on the Varnished Surface Color”.   Some parts of this manuscript overlap exactly with the previous article and duplicate sentences and materials and methods have been used and the similarity should be reduced. 

Author Response

Thank you for Your comments and suggestions. We hope our responses and explanations are complete and acceptable and You will agree with them.

Comments:

  1. The results part of manuscript is still long and should be summarized as indicated before . Some figures should be removed!

Response: We agree with your comment. Some changes made. The figures 1,2,4 and 6 were removed.

  1. Another paper has already been published by the same authors in this journal entitled as” The Impact of Sanding and Thermal Compression of Wood, Varnish Type and Artificial Aging in Indoor Conditions on the Varnished Surface Color”.   Some parts of this manuscript overlap exactly with the previous article and duplicate sentences and materials and methods have been used and the similarity should be reduced.

Response: We agree with your comment. The paragraphs 2.1, 2.2, 2.3 and 2.5 were removed and the next sentence “The content of this paragraph is described with sufficient details in our previous article [36].” added to the text.

Reviewer 3 Report

Accept as it is.

Author Response

Thank you very much for  Your valuable comment and suggestion.